# Models of Risk Selection in Maternal and Newborn Care: Exploring the Organization of Tasks and Responsibilities of Primary Care Midwives and Obstetricians in Risk Selection across The Netherlands

**DOI:** 10.3390/ijerph19031046

**Published:** 2022-01-18

**Authors:** Bahareh Goodarzi, Corine Verhoeven, Durk Berks, Eline F. de Vries, Ank de Jonge

**Affiliations:** 1Department of Midwifery Science, AVAG, Amsterdam Public Health, Amsterdam UMC, Vrije Universiteit Amsterdam, Van der Boechorststraat 7, 1081 BT Amsterdam, The Netherlands; c.verhoeven@amsterdamumc.nl (C.V.); ank.dejonge@amsterdamumc.nl (A.d.J.); 2Queen’s Medical Centre, Division of Midwifery, School of Health Sciences, University of Nottingham, Nottingham NG7 2HA, UK; 3Department of Obstetrics and Gynaecology, Maxima Medical Centre, De Run 4600, 5504 DB Veldhoven, The Netherlands; 4Department of Obstetrics and Gynaecology, Dijklander Ziekenhuis, Maelsonstraat 3, 1624 NP Hoorn, The Netherlands; durk@berks.nl; 5Department of Quality of Care and Health Economics, National Institute for Public Health and the Environment, P.O. Box 1, 3720 MA Bilthoven, The Netherlands; eline.de.vries@rivm.nl; 6Department of Public Health and Primary Care, Campus the Hague, Leiden University Medical Center, 2511 DP Den Haag, The Netherlands

**Keywords:** maternal and newborn care, healthcare organization, risk selection, decision-making, value-based healthcare

## Abstract

An effective system of risk selection is a global necessity to ensure women and children receive appropriate care at the right time and at the right place. To gain more insight into the existing models of risk selection (MRS), we explored the distribution of different MRS across regions in The Netherlands, and examined the relation between MRS and primary care midwives’ and obstetricians’ satisfaction with different MRS. We conducted a nationwide survey amongst all primary midwifery care practices and obstetrics departments. The questionnaire was completed by 312 (55%) primary midwifery care practices and 53 (72%) obstetrics departments. We identified three MRS, which were distributed differently across regions: (1) primary care midwives assess risk and initiate a consultation or transfer of care without discussing this first with the obstetrician, (2) primary care midwives assess risk and make decisions about consultation or transfer of care collaboratively with obstetricians, and (3) models with other characteristics. Across these MRS, variations exist in several aspects, including the routine involvement of the obstetrician in the care of healthy pregnant women. We found no significant difference between MRS and professionals’ level of satisfaction. An evidence- and value-based approach is recommended in the pursuit of the optimal organization of risk selection. This requires further research into associations between MRS and maternal and perinatal outcomes, professional payment methods, resource allocation, and the experiences of women and care professionals.

## 1. Introduction

Globally, high rates of preventable morbidity and mortality persist for childbearing women and their children (when we use the term ‘woman’, we also refer to individuals with a uterus who are also not woman identified, including trans and non-binary individuals) [1,2,3,4]. Preventable morbidity and mortality are associated with both underuse and overuse of care [5,6,7]. To ensure women and children receive appropriate care necessitates a maternal and newborn care (MNC) system focused on promoting health, preventing complications, and warranting timely transition to medical specialist care if needed [2].

Effective risk selection is essential to achieve appropriate MNC [8,9]. Internationally, risk selection is used as an organizational measure to optimally align childbearing women’s needs and MNC resources, ensuring that care is provided by healthcare professionals with the appropriate level of expertise, in the most appropriate place, where the appropriate facilities and resources are located, and with the type and timing of care planned appropriately [10].

In The Netherlands, midwives and obstetricians aim to optimize the organization of tasks and responsibilities in risk selection through an integrated, multidisciplinary approach in order to improve care outcomes. This has opened the door to experiments with different models of risk selection (MRS) [11,12,13,14,15]. Historically, MNC has been organized into primary midwife-led care and obstetrician-led hospital-based care [16,17]. Similar to other countries, midwives and obstetricians use a national multidisciplinary evidence-based list of medical indications for consultation and transfer of care to obstetricians (LMI) [18,19,20,21,22]. This list includes agreements about the division of tasks and responsibilities between midwives and obstetricians regarding risk selection [16,17]. Primary care midwives are the primary caregivers for women with a healthy and uncomplicated pregnancy and birth. As ‘gatekeepers’, they are also responsible for risk selection: they identify women with risk factors or complications and initiate a consultation or transfer of care to obstetrician-led care in the hospital. These women give birth in the hospital under supervision of the obstetrician. When specialist care is no longer required, women are referred back to the primary care midwife, where they can choose to either give birth at home, in a birth center, or in the hospital under supervision of the primary care midwife [16,17]. Over the past several years, in some regions, obstetricians have become routinely involved in risk selection, although their tasks and responsibilities vary considerably. In other regions, primary care midwives’ tasks and responsibilities have been extended, including the assessment of women with high-risk profiles [13,14,15]. Until now, these experiments have not been systematically evaluated, which impedes comparisons of the different MRS.

The evaluation and comparison of MRS across various settings has been identified as one of the top research priorities necessary to improve MNC [23,24]. A recent scoping review into risk selection showed that the organization of risk selection is highly contextualized, determined by numerous factors, including geography, demography, government policy, laws and regulations, history, and culture. The contextual relativity of risk selection is a major challenge for the study of risk selection. The improvement of risk selection can only be achieved through context-specific research with an understanding of the variation in MRS [10]. Additionally, these studies are necessary for an evidence-based reform of MRS. Therefore, we explored the different MRS across The Netherlands in terms of the organization of tasks and responsibilities between primary care midwives and obstetricians and the distribution across regions. We also examined the relation between primary care midwives’ and obstetricians’ level of satisfaction and different MRS.

## 2. Materials and Methods

### 2.1. Design

In this nationwide study, we used a questionnaire to gain insight into the organization of tasks and responsibilities of primary care midwives and obstetricians in the different MRS The Netherlands.

### 2.2. Study Population

In The Netherlands, approximately two-thirds of midwives work in primary midwifery care community-based practices. The remaining one-third practice in the hospital under supervision of an obstetrician and are referred to as hospital-based midwives [25]. Regionally, hospitals and the surrounding primary midwifery care practices are organized into 71 Maternity Care Collaborations (MCC) [26]. MCCs vary in many aspects, including the involvement of other professionals in multidisciplinary discussions—such as pediatricians, general practitioners, maternity care assistants, social workers, and child health nurses—and the way in which discussions between professionals take place, for example, face-to-face, by telephone, by e-mail, or by video call.

### 2.3. Questionnaire

A web-based questionnaire was designed by the members of the research team. The questionnaire consisted of four sections. Section one contained questions about the characteristics of primary midwifery care practices and obstetrics departments. Sections two and three of the questionnaires were based on the LMI [16,17]. We asked questions about the division of tasks and responsibilities between primary care midwives and obstetricians regarding the booking appointment, risk assessment after the booking appointment, whether discussions were routinely scheduled or not, the location and moment of these discussions, which professionals attended these discussions, and the decision-making after these discussions. These questions were specified for five categories of medical risk—uncomplicated, medium-risk, high-risk, unclear risk, and psychosocial risk. We used multiple choice questions and open fields for remarks. The fourth section contained questions about the level of satisfaction with the organization of the risk selection.

We requested and received feedback from the College for Perinatal Care (a collaboration between obstetricians, midwives, pediatricians, maternity care assistants, hospitals, insurance companies, and The Netherlands Patient Federation committed to enhance the integration of maternal and newborn care (https://www.kennisnetgeboortezorg.nl/) access date: 1 October 2021); the Federation of Maternity Care Collaborations (a collaboration between the representatives of MCCs (https://www.federatievsv.nl/) access date: 1 October 2021); the Child and Hospital Foundation (represents children and their families in medical care (https://kindenziekenhuis.nl/) access date: 1 October 2021); The Dutch Society of Obstetrics and Gynecology [27]; the Royal Dutch Organization of Midwives [28]; and an independent advisor in obstetrics and perinatology. The questionnaire was pilot tested in two stages in August and September 2019 by six primary care midwives, four hospital-based midwives, and two research assistants. The pilot test resulted in minor adjustments to the questionnaire, including clarification of the categories of medium-risk and high-risk, changes in the ranges for the number of care, and number of births and the need to purposively invite participants with knowledge about the agreements about the risk selection processes.

### 2.4. Data Collection

Data was collected in the period October 2019 to September 2020 using Castor EDC software (2019) [29]. We invited all primary midwifery care practices and obstetrics departments across the 71 MCCs in The Netherlands to participate in this study. Invitations were sent by e-mail in October 2019. At the time of the study, an up-to-date overview of primary midwifery care practices did not exist. Therefore, we used an overview provided by the three Midwifery Academies, the Perined data register (the Dutch national perinatal register, which includes data on pregnancies and births collected from primary care midwives, obstetricians, and pediatricians (https://www.perined.nl/) access date: 1 October 2021), and the College for Perinatal Care. Missing e-mail addresses of primary midwifery care practices and obstetrics departments were obtained from the internet or by phone.

To make sure we did not miss any primary midwifery care practices and obstetrics departments, and to achieve a high response rate, we announced the study in widely read media amongst midwives and obstetricians, including the College for Perinatal Care’s newsletter [30], the Dutch journal for midwives [31], the Dutch journal for obstetricians and gynecologists [32], Kennispoort Verloskunde [33], and social media.

The study invitation, containing information about the study and a link to the questionnaire, was sent to 569 (99.8%) primary midwifery care practices and 74 (100%) obstetrics departments. We invited professionals in each primary midwifery care practice and obstetrics department to fill in the questionnaire once on behalf of their practice or department. Via the link to the questionnaire, professionals could accept or decline participation. Those who accepted participation were forwarded to the questionnaire. The primary midwifery care practices and obstetrics departments that did not respond to the invitation received a reminder e-mail in February 2020. In the first section of the questionnaire, we asked what practices and departments of which MCC they were a member and to list the other primary midwifery care practices that were a member of their MCC. In the case of membership to multiple MCCs, respondents were asked to answer the questionnaire with regards to the MCC with which they collaborated most. We used these answers to group the primary midwifery care practices and obstetrics departments into MCCs. The data was anonymized and stored in a secured, password-protected digital storage system at the Department of Midwifery Science at the Vrije Universiteit Amsterdam.

### 2.5. Analyses

First, we grouped the respondents at the level of MCCs using Excel software [34]. We combined the answers of the primary midwifery care practices and obstetrics departments that were members of the same MCC. The answers that were inconsistent or unclear were categorized in three categories: ‘contradictory’, ‘variation within the MCC’, and ‘unclear’. The category ‘contradictory’ was used if the answers among the respondents within a MCC contradicted. The category ‘variation within the MCC’ was used if members of the same MCC indicated that a variation in the division of tasks and responsibilities within the MCC existed. We used the category ‘unclear’ if the combined answers of the respondents were unclear. Then, we identified different MRS by grouping the different MCCs with the same division of tasks and responsibilities between primary care midwives and obstetricians. The answers about the level of satisfaction were combined in three ways: by calculating the average score of the primary midwifery care practices belonging to the same MCC, the average score of the obstetrics departments belonging to the same MCC, and the average score of both.

We grouped the data at the level of MCCs in a two-stage process to minimize the bias. First, the researchers BG and CV independently categorized the data of the first five MCCs. This resulted in some disagreements about how to categorize contradictory and unclear answers and answers indicating different policies within an MCC. After BG and CV reached consensus on the classification, they independently combined the data of another five MCCs, resulting in no differences. The remaining data were categorized by BG.

The data were analyzed using STATA software [35]. We used descriptive statistics (*n*,%) to report the characteristics of the MCCs, the organization of tasks and responsibilities of primary care midwives and obstetricians within MCCs, the distribution of MRS across regions, and the level of satisfaction of MCCs. To examine the distribution of different MRS across regions, the MCCs were divided into five regions: ‘north’, ‘east’, ‘south’, ‘southwest’ and ‘northwest’ [36,37]. The level of satisfaction was categorized in binary categories based on the mean score for level of satisfaction for primary care midwives within a MCC, obstetricians within an MCC, and the overall score of an MCC. We combined the categories ‘very satisfied’ and ‘satisfied’ and the categories ‘a little satisfied’, ‘neutral’, ‘a little unsatisfied’, ‘unsatisfied’, and ‘very unsatisfied’ into the categories ‘very satisfied’ and ‘not very satisfied’ [38]. The chi-square test of independence and the Fisher’s exact test were calculated to examine the relation between the MRS and level of satisfaction. *p*-values < 0.05 were considered significant.

## 3. Results

### 3.1. Characteristics of the Included Primary Midwifery Care Practices and Obstetrics Departments

The questionnaire was completed by 312 (55%) primary midwifery care practices and 53 (72%) obstetrics departments. Participation was declined by 68 (12%) primary midwifery care practices and seven (9%) obstetrics departments (Figure 1). To check the national representation in our study sample, in June 2020, we geographically mapped the primary midwifery care practices and obstetrics departments that responded. We found an unequal representation of obstetrics departments. Therefore, in September 2020, all nonresponding obstetrics departments received a reminder phone call. The characteristics of the included primary midwifery care practices and obstetrics departments are presented in Table 1.

We received a response from 69 out of the 71 MCCs (96%). In four (6%) MCCs, we received a response from all primary midwifery care practices and obstetrics departments that were members of that MCC. In 17 of the 69 MCCs (25%), we received no response from the obstetrics departments. In two (3%) MCCs, we received no response from primary midwifery care practices. The size of the primary midwifery care practices and obstetrics departments ranged 30–2000 care units and 650–4000 births per year, respectively. Forty-three (14%) primary midwifery care practices reported being a member of two MCCs, and five primary midwifery care practices (2%) reported being member of three MCCs. No obstetrics departments reported being member of more than one MCC. In eight (12%) MCCs, one or two primary midwifery care practices reported the membership of an MCC, that was not confirmed by the obstetrics departments. Sixty-one (20%) primary midwifery care practices reported collaborations with a maximum of four hospitals outside their MCC.

### 3.2. Characteristics of Models of Risk Selection

The results of the organization of the tasks and responsibilities at the level of MCCs are shown in Table 2, Table 3 and Table 4. Table 2 shows that, in most MCCs, all women with uncomplicated pregnancies started their care in primary midwifery care practices, and women with a high-risk profile or with existing complications started their care in secondary obstetrician-led care in the hospital. Primary care midwives assessed the risk and initiated a consultation or transfer of care only if necessary, without discussing this first with the obstetrician. Bi-disciplinary discussions between primary care midwives and obstetricians were sometimes scheduled upon request and sometimes scheduled routinely. After discussion, primary care midwives and obstetricians together made decisions regarding a consultation or transfer of care.

In 75% of the MCCs, the moments of discussion between primary care midwives and obstetricians were routinely scheduled (Table 3). Meetings were scheduled at a certain moment in the care pathway—mostly after the booking appointment (33%)—or with a certain frequency, mostly every six weeks (11%), every two weeks (14%), or weekly (11%). In almost half of the MCCs, the bi-disciplinary discussions were also attended by hospital-based midwives. In 22% of the MCCs, resident obstetricians were also part of these discussions (Appendix A). In some of the MCCs, one or more visits were scheduled routinely for women in primary midwifery care to the obstetrician and vice versa (Table 4).

Multidisciplinary discussions were scheduled routinely in half of the MCCs (Table 3). Of these, 89% were for women with psychosocial complications. In four (11%) MCCs, all indications requiring a multidisciplinary approach were discussed. In half of the MCCs, the multidisciplinary discussions were scheduled halfway through pregnancy. The professionals most often present at the multidisciplinary discussions were the primary care midwife (95%), the obstetrician (92%), the pediatrician (89%), the social worker (78%), and the child health nurse (49%). In some MCCs, the general practitioner, the psychiatrist, and a representative of the child protection council also attended the multidisciplinary discussions. In one MCC, other professionals only attended if relevant for the discussed case. In three MCCs, other professionals were always present; in one of these MCCs, all booking appointments conducted in primary midwifery care practices were discussed multidisciplinary. In four MCCs, the scheduled discussions between primary care midwives and obstetricians and the multidisciplinary discussions were combined. Eighty-one percent of the multidisciplinary discussions took place face-to-face in the hospital and 47% by phone (Appendix A). All MCCs reported that decisions were made after discussion.

### 3.3. Models of Risk Selection

Based on variations in the organization of tasks and responsibilities (Table 2, Table 3 and Table 4), we identified three MRS. In all models, primary care midwives were responsible for the booking appointment of women with uncomplicated pregnancies. The models differed in the organization of tasks and responsibilities for risk assessment after the booking appointment about consultation or transfer of care. Table 5 shows the prevalence of the models of care by region.

Model 1: Primary care midwives assessed the risk and initiated a consultation or transfer of care after the booking appointment only if necessary, without discussing this first with the obstetrician. This model resembled the usual MRS and was used in 61% (42) MCCs, which were mainly located in the northwest of The Netherlands.

Model 2: Risk was assessed collaboratively after discussion between the primary care midwives and obstetricians. This model was used by 23% (16) of MCCs, which were mainly located in the south of The Netherlands. In 20% (14) of the MCCs, primary care midwives and obstetricians discussed all women who had had a booking appointment with the primary care midwife and whether consultation or transfer of care was indicated. In 2% (three) of the MCCs, women who had their booking appointment in obstetrician-led care were also discussed.

Model 3: In 16% (11) of the MCCs, the organization of the tasks and responsibilities regarding risk assessment after the booking appointment within a MCC was unclear or varied amongst the primary midwifery care practices and obstetrics departments that were members of an MCC. This model was predominantly prevalent in the north of The Netherlands.

In both model 1 and 2, more than half of the MCCs indicated shared decision-making by primary care midwives and obstetricians. The third model consisted of MCCs where the MRS remained unclear due to contradictory answers amongst respondents within a MCC or variations in the division of tasks and responsibilities within the MCC.

### 3.4. Change in the Usual Organization of Tasks and Responsibilities in Risk Selection

Seventy-two percent (50) of the MCCs made changes in their usual organization of tasks and responsibilities in the risk selection between 2011 and 2019 (Table 6). In 28 (61%) of these MCC, these changes were made between 2014 and 2016. In nine (18%) of these MCCs, the primary care midwives became responsible for all booking appointments, including the booking appointments of women with high-risk profiles. In 21 (42%) of these MCCs, the primary care midwives and obstetricians together started discussing all women who had a booking appointment with a primary care midwife, of which two MCCs also discussed all women who had a booking appointment in obstetrician-led care. Forty-eight (96%) MCCs started routinely planning discussions between midwives and obstetricians, and six (12%) MCCs started routinely planning multidisciplinary discussions. Two (4%) of these MCCs started with routine visits for all women to the obstetrician.

Nine (18%) MCCs of the 50 MCCs that made changes in their usual organization of tasks and responsibilities in risk selection changed back to their usual organization of tasks and responsibilities in risk selection (Table 6). Five (10%) changed back to their usual organization of tasks and responsibilities in risk selection because of time constraints or the loss of autonomy of the primary care midwives. Four (8%) MCCs changed some of the characteristics of the newly implemented MRS back to their usual organization of tasks and responsibilities in risk selection because of time constraints, financial constraints, or because the changes were experienced as unnecessary (Appendix A).

### 3.5. Level of Satisfaction

The results of the descriptive analysis of level of satisfaction for the three MRS are presented in Appendix A. Overall, the respondents indicated being very satisfied. The respondents were least satisfied about their time investment (73%). Respondents from primary midwifery care practices were more often very satisfied about the quality of care (84% versus 69%), and respondents from obstetrics departments were more often very satisfied about time investment (53% versus 20%) and autonomy (81% versus 73%). Primary care midwives working in MCCs using MRS 3 were least satisfied, except for the quality of collaboration and organization of care, where those in MRS 1 and model 2 were least satisfied, respectively. Primary care midwives working in MCCs using MRS 2 were most satisfied, except for autonomy and organization of care, where those in MRS 1 and model 3 were least satisfied, respectively. Respondents from the obstetrics departments indicated the least satisfaction with the quality of care and quality of collaboration when working in MCCs using in MRS 1 and time investment and autonomy when working in MCCS using MRS 3. Respondents from obstetrics departments working in MCCs using MRS 2 indicated the most satisfaction, except for the organization of care, which the obstetrics departments working in MCCs using MRS 3 were most satisfied about.

The results of the chi-square test are shown in Table 7. We found no significant difference between the MRS and levels of satisfaction. Tests for primary midwifery care practices and obstetrics departments only showed statistically significant differences for the relation between quality of care and respondents from obstetrics departments (*p* = 0.037, MRS 1 very satisfied 17 (47%), ‘a little satisfied’ 13 (81%), model 2 ‘very satisfied’ 13 (26%), ‘a little satisfied’ 1 (6%), model 3 ‘very satisfied’ 6 (17%), and ‘a little satisfied’ 2 (13%)).

## 4. Discussion

Exploring the organization of tasks and responsibilities of primary care midwives and obstetricians in risk selection in The Netherlands, we identified three MRS. We found that the majority of the MCCs work according to the usual MRS, where primary care midwives assess the risk, and initiate a consultation or transfer of care after the booking appointment only if necessary, without discussing this first with the obstetrician. In the second MRS, after the booking an appointment, primary care midwives assess the risk and make decisions about the consultation or transfer of care collaboratively with obstetricians. The third model consists of models with other characteristics. We did not find significant differences between the MRS and levels of satisfaction. 

Our survey showed that most MCCs in The Netherlands work according to the usual MRS, where primary care midwives initiate a consultation or transfer of care only if necessary, without discussing this first with an obstetrician, and conform to the agreements laid down in the division of tasks and responsibilities between midwives and obstetricians in the LMI [13,14,15]. Experiments in the past decade with MRS included a model in which midwives and obstetricians always collaboratively assess risks and make decisions, which is now used by almost a quarter of the MCCs [13,14,15]. The MRS vary in several aspects, including routinely scheduled discussions, attending professionals at discussions, and routine visits to the obstetrician. The results of this study can inform other healthcare systems about how risk selection can be organized in different ways.

The experiments with MRS were not evidence-based but were prompted by a consensus-based report in 2009, with recommendations to improve the MNC outcomes, such as woman-centered care, preventive care, and accessible care [39]. A specific recommendation was to improve the shared responsibility by a multidisciplinary discussion of all women in a MCC [39] (pp. 32 & 72). Post-humus and colleagues (2013) explored barriers for shared responsibility, including negative financial incentives and a lack of mutual respect [40]. According to them, the usual MRS and the LMI were unique features of the Dutch MNC system and key barriers to shared responsibility, leading to avoidable perinatal mortality [40,41,42]. However, having agreed guidelines for consultation and transfer of care has been identified as an essential component of successful collaboration in MNC in the international literature [10,43]. Primary care midwives and obstetricians in many other countries use an LMI for consultation and referral to organize tasks and responsibilities, including Canada [22], New Zealand [18], Australia [19], South Africa [20], and England [21]. Belgium is currently developing such a list to support the collaboration between primary care midwives and obstetricians, because an increasing number of women are choosing primary midwifery care [44].

Since the start of the experiments with MRS, only two small experiments were evaluated. Both evaluations used a retrospective cohort design comparing an intervention and a control group. In one experiment, midwives and obstetricians discussed all women who had their booking appointment in primary midwifery care practices. No changes were found in the rates of consultation and transfer of care [13]. The other experiment also included women who had their booking appointment in obstetrician-led care. No changes were found in substitution of care towards primary midwife-led practices and cost reduction [14]. Both experiments did not have the power to evaluate the medical intervention rates and maternal and perinatal morbidity and mortality rates. In both experiments, the women and care providers were satisfied. However, the level of satisfaction was only measured in the intervention group.

In this study, we did not find significant differences between the MRS and levels of satisfaction. Previous studies showed that midwives and obstetricians were positive about intensified collaboration. They indicated well-defined responsibilities between midwives and obstetricians, individual responsibility, and collaborative discussions of all women as facilitators of integrated care [45,46,47]. However, on the one hand, they expressed a preference to remain autonomous in decision-making and organization of care [46,47], while, on the other hand, they considered autonomy as a barrier to integrated care [45]. The three MRS identified in this study differed in the degree of professional individual autonomy, particularly that of primary care midwives. Further in-depth research is needed to better understand the association between level of autonomy and level of satisfaction.

In many of the MCCs, healthy women in primary midwifery care were discussed with the obstetrician, or visited the obstetrician at least once during the prenatal period. To our knowledge, the benefits of routine visits to the obstetrician of healthy women with uncomplicated pregnancies in midwife-led models of care remain unknown. Arguably, routine involvement of obstetricians in all healthy pregnancies enhances the continuity of care, because many women in primary midwifery care will eventually be referred to obstetrician-led care at some point during their care [48]. However, routine visits do not necessarily contribute to relational continuity, because the care providers that women see during the visits are often not the same care providers they will see during consultations or birth [49,50]. Informational continuity can be achieved with an electronic medical record accessible by all professionals involved in the care [49]. The continuity of case management is important for women with complex complications, which is facilitated by shared agreements and flexibility in care provision [49].

Lastly, our results indicate that, even though the experiments with MRS were motivated by the desire to improve quality of care [11,12,13,14], in some MCCs, factors such as a lack of time and finances were reasons to reverse changes in the organization of tasks and responsibilities in risk selection. Other studies into facilitators and barriers of collaboration between midwives and obstetricians also indicate a willingness to invest time as a facilitator of collaboration and a lack of finance as a barrier to collaboration and integrated care [45,46,51,52]. The current MNC payment system in The Netherlands may encourage professionals’ financial interest and reduce their willingness to invest time, which may influence decisions regarding MRS. The payment of midwives and obstetricians differs. Obstetricians working in university hospitals, and hospital-based midwives generally work as employees. Primary care midwives and obstetricians working in regional hospitals are generally self-employed and are paid on a fee-for-service basis. Some MCCs are experimenting with bundled payments. It remains unclear with payment system facilitates optimal MRS the most [53,54].

The results of this study call for an evidence- and value-based approach. The insights of this study into different MRS are necessary to further study the association between MRS and care outcomes. Systematic evaluations using randomized controlled trial or quasi-experimental designs are needed to study the association between MRS and outcomes. These evaluations should include effectiveness studies into allocation of resources, such as invested time and financial availability. Additionally, qualitative research is required to gain a better understanding of the association between professionals’ payment methods and decisions about MRS and the experiences of care professionals and women and different MRS. These studies can inform a high-value reform, achieving the best possible health outcomes with an optimal use of resources [8,55,56,57].

## 5. Strengths and Limitations

A strength of this study is that it is the first to explore the organization of tasks and responsibilities of primary care midwives and obstetricians in risk selection across The Netherlands. We used a questionnaire about the objective characteristics MRS to gain insight into the complexity of the organization of risk selection. The questionnaire was reviewed by many stakeholders and pilot-tested. We received responses from 69 out of the 71 MCCs in The Netherlands, meaning that almost all MCCs were represented in our sample. We received two or more responses in 94% of the MCCs, which provided us with multiple perspectives of the organization of tasks and responsibilities. However, a limitation of the study is that only representatives of the primary midwifery care practices and obstetrics departments were invited to fill in the questionnaire on behalf of their practice or department; they may not have represented the views and experiences of the whole practice or department. Additionally, in almost one-third of the MCCs, the participants’ responses of either the primary midwifery care practices or the obstetrics department were missing, which might have resulted in a partial view of the organization of tasks and responsibilities within a MCC [58,59].

## 6. Conclusions

In this study, we identified three MRS across regions in The Netherlands: (1) primary care midwives assess risk and initiating a consultation or transfer of care without discussing this first with the obstetrician, (2) primary care midwives assess the risk and make decisions about the consultation or transfer of care collaboratively with the obstetrician, and (3) models with other characteristics. Across these MRS, variations exist in several aspects, including the routine involvement of the obstetrician in the care of healthy pregnant women. We found no significant association between the MRS and levels of satisfaction. We recommend an evidence- and value-based approach in the pursuit of the optimal organization of risk selection. Further research is needed to analyze the associations between different MRS and maternal and perinatal outcomes, professional payment methods, resource allocation, and the experiences of women and care professionals.

## Figures and Tables

**Figure 1 ijerph-19-01046-f001:**
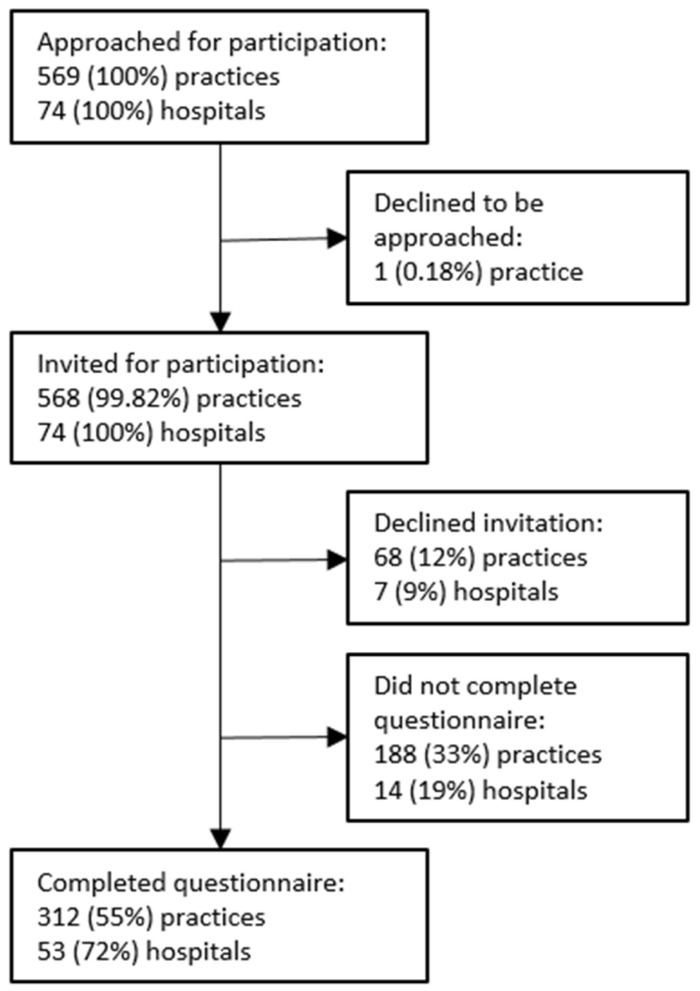
Study population.

**Table 1 ijerph-19-01046-t001:** Characteristics of the responding primary midwifery care practices and obstetrics departments (*n*,%).

	Primary Midwifery Care Practices		Obstetrics Departments
	*n* (%)		*n* (%)
Total	312 (100)	Total	53 (100)
Profession		Profession	
Primary care midwife	312 (100)	Hospital-based midwife	9 (17)
		Midwife working both as primary care midwife and hospital-based midwife	1 (2)
		Obstetrician	30 (57)
		Manager	13 (25)
Number of care units/year ^1^		Number of births/year ^2^	
≤250	181 (58)	≤1000	10 (19)
251–500	117 (38)	1001–2000	21 (40)
501–750	11 (4)	2001–3000	15 (21)
≥751	3 (1)	≥300–4000	6 (12)

^1^ Care by primary care midwives in The Netherlands are reimbursed on an episode-based payment basis of the average costs of prenatal, natal, and postnatal care of one woman. Missing: 2 (0.6). ^2^ Care by obstetricians in The Netherlands is reimbursed on a fee for service basis. Missing: 1 (0.3).

**Table 2 ijerph-19-01046-t002:** The booking appointment, risk assessment after booking appointment, and decision-making after discussion between primary care midwives and obstetricians (*n*,%).

The Organization of Tasks and Responsibilities Regarding the Booking Appointment	*n* (%)
All women start their care in primary midwifery care practices, regardless of their risk profile. Primary care midwives are responsible for the booking appointment.	11 (16)
All women start their care in secondary obstetrician-led care in the hospital, regardless of their risk profile. Obstetricians are responsible for the booking appointment.	0 (0)
All women with uncomplicated pregnancies start their care in primary midwifery care, and women with a high-risk profile start their care in secondary obstetrician-led care in the hospital. Primary care midwives are responsible for the booking appointment for women with uncomplicated pregnancies and obstetricians are responsible for the booking appointment of women with a high-risk profile.	52 (75)
Contradictory answers amongst respondents within the MCC *.	6 (9)
Organization of tasks and responsibilities regarding risk assessment after booking appointment.	
Primary care midwives assess risk and initiate a consultation or transfer of care only if necessary, without discussing this first with the obstetrician.	42 (61)
Risk is assessed collaboratively. Primary care midwives and obstetricians discuss whether a consultation or transfer of care is necessary: - Only for women who had a booking appointment with a primary care midwife; - All women.	16 (23) - 14 (20) - 2 (3)
Contradictory answers amongst respondents within the MCC.	4 (6)
Variation within the MCC.	3 (4)
Unclear.	4 (6)
Organization of tasks and responsibilities in decision-making after discussion between primary care midwives and obstetricians.	
Primary care midwives and obstetricians are together responsible for decision-making.	100 (100)
Primary care midwives are responsible for decision-making.	0 (0)
Obstetricians are responsible for decision-making.	0 (0)

* MCC = Maternity Care Collaboration.

**Table 3 ijerph-19-01046-t003:** Moment of discussion specified for bi-disciplinary and multidisciplinary * discussions (*n*,%).

	Bi-Disciplinary	Multidisciplinary
	*n* (%)	*n* (%)
Discussion is only scheduled at request, and not scheduled routinely.	9 (13)	28 (40)
Discussion is scheduled at request, and scheduled routinely.	52 (75)	37 (54)
Contradictory answers amongst respondents within the MCC *.	1 (1)	2 (3)
Variation within the MCC.	6 (9)	0
Unclear.	1 (1)	2 (3)

* Bi-disciplinary discussions: discussion between primary care midwives and obstetricians. Multidisciplinary discussions: discussion between primary care midwives and obstetricians and other professionals.

**Table 4 ijerph-19-01046-t004:** Routinely scheduled visit to the obstetrician or the primary care midwife (*n*,%).

	*n* (%)
One or more visits to the obstetrician are scheduled routinely for all women in primary midwife-led care.	1 (1)
One or more visits to the primary care midwife are scheduled routinely for all women in obstetrician-led care.	3 (4)
One or more visits to the obstetrician are scheduled routinely for women with a medium or high-risk profile in primary midwife-led care.	11 (16)
One or more visits to the primary care midwife are scheduled routinely for all women in obstetrician-led care, and vice versa to the obstetrician for all women in primary midwife-led care.	2 (3)
One or more visits to the primary care midwife are scheduled routinely for women in obstetrician-led care, and vice versa to the obstetrician for women with a medium or high-risk profile in primary midwife-led care.	1 (1)
Variation within the MCC *.	1 (1)
Unclear.	4 (6)
Not mentioned in the answers.	46 (67)

* MCC = Maternity Care Collaboration.

**Table 5 ijerph-19-01046-t005:** Models of risk selection specified for region (*n*,%).

	MRS * 1	MRS 2	MRS 3
	*n* (Row %)	*n* (Row %)	*n* (Row %)
Total	42 (61)	16 (23)	11 (16)
North	3 (33)	1 (11)	5 (56)
East	9 (64)	4 (29)	1 (7)
South	2 (14)	10 (71)	2 (14)
Southwest	12 (75)	1 (6)	3 (19)
Northwest	16 (100)	0 (0)	0 (0)

* MRS = Model of risk selection.

**Table 6 ijerph-19-01046-t006:** Change in the usual organization of tasks and responsibilities in risk selection (*n*,%).

	Change in the Past Decade	Reversed Change Back to the Usual Model
	*n* (%)	*n* (%)
Total MCCs ^#^	69 (100)	50 (100) *
No	15 (22)	41 (82)
Yes	46 (67)	9 (18)
Variation within the MCC	4 (5)	
Contradictory answers amongst respondents within the MCC	4 (5)	

* Sum of ‘yes’ and ‘variations within the MCC’. ^#^ MCC = Maternity Care Collaboration.

**Table 7 ijerph-19-01046-t007:** Level of satisfaction by Maternity Care Collaboration, primary midwifery care practices and obstetrics departments (*p*).

	MCC ^#^		Primary Midwifery Care Practices		Obstetrics Departments	
	Very Satisfied	Not Very Satisfied	Very Satisfied	Not Very Satisfied	Very Satisfied	Not Very Satisfied
	*n* (%)	*n* (%)	*n* (%)	*n* (%)	*n* (%)	*n* (%)
Quality of care						
MRS ^#^ 1	32 (57)	10 (77)	34 (61)	7 (63)	17 (47)	13 (81)
MRS 2	14 (25)	2 (15)	13 (23)	2 (18)	13 (36)	1 (6)
MRS 3	10 (18)	1 (7)	9 (16)	2 (18)	6 (17)	2 (13)
Collaboration						
MRS 1	28 (55)	14 (78)	26 (54)	15 (79)	20 (53)	10 (71)
MRS 2	13 (25)	3 (17)	13 (27)	2 (12)	12 (32)	2 (14)
MRS 3	10 (20)	1 (6)	9 (19)	2 (12)	6 (16)	2 (14)
*p*-value *	0.23		0.17		0.46	
Organization of care						
MRS 1	20 (56)	22 (67)	22 (63)	19 (59)	15 (54)	15 (63)
MRS 2	9 (25)	7 (21)	7 (20)	8 (25)	8 (29)	6 (25)
MRS 3	7 (19)	4 (12)	6 (17)	5 (16)	5 (18)	3 (13)
*p*-value *	0.66		0.94		0.80	
Time investment						
MRS 1	12 (67)	30 (59)	9 (64)	32 (60)	14 (64)	16 (53)
MRS 2	3 (17)	13 (25)	3 (21)	12 (23)	5 (23)	9 (30)
MRS 3	3 (17)	8 (16)	2 (14)	9 (17)	3 (14)	5 (17)
*p*-value *	0.74		1.00		0.74	
Autonomy						
MRS 1	32 (64)	10 (53)	32 (65)	9 (50)	23 (55)	7 (70)
MRS 2	11 (22)	5 (26)	11 (22)	4 (22)	11 (26)	3 (30)
MRS 3	7 (14)	4 (21)	6 (12)	5 (28)	8 (19)	0
*p*-value *	0.64		0.29		0.44	

* Fisher exact test, statistically significant (*p* < 0.05). ^#^ MCC = Maternity Care Collaboration. ^#^ MRS = models of risk selection.

## Data Availability

Datasets were not publicly archived datasets analyzed or generated during the study.

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
