# Peer review of "Models of Risk Selection in Maternal and Newborn Care: Exploring the Organization of Tasks and Responsibilities of Primary Care Midwives and Obstetricians in Risk Selection across The Netherlands"

_ijerph, 2022, doi:10.3390/ijerph19031046_

Round 1

Reviewer 1 Report

In this work of Bahareh Goodarzi et al, the authors perform a nationwide questionnaire to survey model of risks in maternal and newborn care. Though the study was almost descriptive,  the data remain very important since the response rate was high .

This paper may be an important reference for countries having simultaneously midwives’ and obstetricians' systems. Helping midwives practice to initiate and transfer high risk pregnancy to obstetrician office does improve the better perinatal care and reduce the load of obstetrician.

The weak point was the responders were the representatives of all primary midwifery care practices and obstetrics departments. Some tables could be improved by reducing the wording and the table layout should be further polished.  

Reviewer 2 Report

  1. Summary for manuscript IJERPH-1474084: Models of risk selection in maternal and newborn care: exploring the organization of tasks and responsibilities of primary care midwives and obstetricians in risk selection across the Netherlands. This is nationwide descriptive survey designed to evaluate provider satisfaction with 3 models of risk selection for pregnant individuals. The response rate was 55% (312) for midwife practices and 72% (53) for obstetrics departments. There were no differences with MRS satisfaction among providers.

  2. The strengths of the manuscript are: A nationwide survey to evaluate the model used for patient referral between primary care midwife providers and hospital centered care by obstetricians or midwives. The authors obtained responses from 55% of th4 primary care centers and 72% of the obstetrical service centers.

Additional strengths are: having a clear explanation of how care providers are organized in the Netherlands, the existence of a list of medical indications for transfer of care, having identified that most transfer discussions are related to psychosocial complications of pregnancy, and having identified how the decision for referral is made, either at first visit for prenatal care, or at the time of identifying a complication. The decision is made by the primary care provider, by the obstetrician, by both or by a combination of these.

Limitations are: There is no outcome data to evaluate the effectiveness/cost ratio or value of the different forms of interaction which are classified according to the degree of co-decision making between primary care and hospital centered providers when deciding on a transfer of care.

The relationship will depend on the value of the interaction to all interested parties: obstetricians, midwives, patients and payers.

One important aspect of this study that was not clear is where births occur and who is caring for individuals at the time of the delivery. If all deliveries in Holland are hospital based, then the referral to a hospital based practice is inevitable, if hospital based deliveries can be performed by either a midwife or an obstetrician, referral could occur to either a midwife or obstetrician. If place of delivery is based exclusively on risk and referral occurs based on risk, this assumes that it is in the interest of payers, patients and providers that this occur consistently based on a set of conditions or comorbidities that can be diagnosed early or later in the pregnancy. If this is the case, the mode of interaction  (MRS) will depend on the method that provides the best value, which is not the same as saying it has no value, regardless of how this is perceived by the providers. An important driver of this discussion will be outcomes and adverse events related to differences in the MRS. It would seem that if there is a specific set of conditions that warrant referral, there is limited option for discussion, if there is a rare condition or that does not fit into any specific referral diagnosis (psychosocial complications), this would require multidisciplinary discussion.

There are minor grammatical errors throughout the manuscript.

  1. As presented, this information has limited clinical importance due to the lack of outcomes. It may guide other healthcare systems in regard to how the decisions to proceed with referrals can occur in various ways, but there is no information to say that one is more effective or valuable than another.

  2. The title adequately conveys the content.
  3. The abstract reflects the content and can be understood without reading the manuscript. There are no discrepancies between the abstract/summary and the remainder of the manuscript.
  4. The introduction succinctly lays the groundwork for what was done.
  5. Materials and Methods: The aims are clearly stated however there is no hypothesis as this is a descriptive study based on a survey. The authors have adequately described the study design and methods. The relatively low response rate from the midwife practices is a limitation.
  6. Results: the results are valid based on the methods, the order of presentation of the results parallel the order of presentation of the methods and no results introduced that are not preceded by an appropriate discussion in the methods.
  7. The discussion adequately compares the results with those of other papers that have previously been published. There is a paragraph that acknowledges study limitations. The discussion does refer to one prior publication reporting differences in financial incentive and respect for the various members of the health team. These are peripheral to the main drivers of the referral systems which have not been addressed in the discussion. The relationship for referral will depend on the value of the interaction to all interested parties: obstetricians, midwives, patients and payers.
  8. The conclusion is rather vague, and although this is a good descriptive study, might benefit from the need to recommend standards for referral as well as the need for outcome data.
  9. Figures: no comment

  10. Tables: no comment
    13. References: are current and comprehensive.

Reviewer 3 Report

The Purpose of the study is a good one for those interested in quality of care. The results presentation needs more thought. I have gotten lost from time to time.
